# Sézary Syndrome in West Sweden: Exploring Epidemiology, Clinical Features, and Treatment Patterns in a Registry-Based Retrospective Analysis

**DOI:** 10.3390/cancers16111948

**Published:** 2024-05-21

**Authors:** Karolina Wojewoda, Martin Gillstedt, Catharina Lewerin, Amra Osmancevic

**Affiliations:** 1Department of Dermatology and Venereology, Institute of Clinical Sciences, Sahlgrenska Academy, University of Gothenburg, 413 46 Gothenburg, Sweden; martin.gillstedt@vgregion.se (M.G.); amra.osmancevic@vgregion.se (A.O.); 2Region Västra Götaland, Department of Dermatology and Venereology, Sahlgrenska University Hospital, 413 46 Gothenburg, Sweden; 3Section of Hematology and Coagulation, Department of Internal Medicine, Institute of Medicine, Sahlgrenska Academy, University of Gothenburg, 413 46 Gothenburg, Sweden; catharina.lewerin@vgregion.se

**Keywords:** Sézary syndrome, cutaneous T-cell lymphoma, primary cutaneous lymphoma, time to next treatment (TTNT), extracorporeal photopheresis (ECP), combination therapy

## Abstract

**Simple Summary:**

Sézary syndrome (SS) is a rare and aggressive form of cutaneous T-cell lymphoma. Despite various treatments, it remains incurable, with a median survival of only 2.1 years. This retrospective study of 17 SS patients in West Sweden from 2012 to 2024 aimed to understand demographic characteristics, treatment effectiveness, and disease progression. Only 35% of patients showed the classic symptoms at diagnosis, indicating the need for personalized diagnostic approaches. Different treatment modalities were used, but combination therapy showed advantages in median survival over monotherapies. Notably, triple therapy involving retinoids, interferon alpha, and extracorporeal photopheresis (ECP) exhibited the longest median time to the next treatment, at 14.1 months. However, early initiation of ECP did not improve outcomes. This study underscores the complexity of SS, emphasizes the urgent need for more effective treatments, and highlights the importance of future prospective research in optimizing treatment strategies.

**Abstract:**

Sézary syndrome (SS) is a rare primary cutaneous T-cell lymphoma variant. Despite various treatment options, it remains incurable, with a poor prognosis. There is an urgent need for additional descriptive research to enhance our understanding and treatment of SS. The aim of this retrospective register-based study was to outline patients’ demographic characteristics; investigate the clinical, histopathological, and molecular findings; and assess treatment effectiveness with a focus on time to next treatment (TTNT) and disease progression. Data on 17 patients with SS were obtained from the primary cutaneous lymphoma register in West Sweden between 2012 and 2024. The results revealed that not all patients exhibited the classical triad of symptoms at diagnosis, emphasizing the need for personalized diagnostic approaches. The median survival was only 2.1 years, which reflects the aggressive nature of SS. The longest median TTNT was observed in triple therapy involving retinoids, interferon alpha, and extracorporeal photopheresis (ECP). There was no significant difference in TTNT between various lines of treatment. Early initiation of ECP treatment did not result in improved outcomes. This study highlights the importance of combination therapy for improved outcomes and underscores the need for future studies to identify optimal treatment approaches.

## 1. Introduction

Cutaneous T-cell lymphomas (CTCLs) constitute approximately 75–80% of all primary cutaneous lymphomas (PCLs) [1]. The different CTCLs comprise a heterogeneous group with various clinical, molecular, and histopathological features. The most common variant of CTCL is mycosis fungoides (MF). Sézary syndrome (SS) constitutes around 3% of all CTCLs, with very little variation between geographical regions except for a slightly higher frequency in Europe [2]. The incidence of SS in both the United States and Europe is 0.1–0.75 per 1,000,000 persons [3,4,5,6]. Median overall survival rates for patients with SS range between 2 and 5 years [7,8,9]. 

There is a triad of features commonly seen in patients with SS: exfoliative or pruritic erythroderma, lymphadenopathy, and neoplastic T-cells in the skin, lymph nodes, and peripheral blood [1]. The erythroderma in SS patients usually emerges de novo, without any other pre-existing skin symptoms. It is important to note that not all patients with SS demonstrate this triad of features, which complicates the diagnostic profile. Other common symptoms include intense prurigo, keratoderma, onychodystrophy, alopecia, leonine facies, and ectropion [10]. Diagnosis of SS requires clinical features in the skin, histopathological examination including molecular analysis, and B2 blood involvement (=≥1000/μL of CD4^+^/CD26^−^ or CD4^+^/CD7^−^ cells) including matched clones in the blood and skin [8,11,12,13].

SS follows a more aggressive course than other variants of CTCLs, and it is also incurable. Fortunately, there are a number of treatment options that can ameliorate symptoms and stop disease progression, though many of these yield only partial responses (PRs) of brief duration [14]. Skin colonization with *Staphylococcus aureus* can contribute to disease flares and has been described in patients with SS [15,16,17,18,19,20]. These difficulties make it crucial to understand which treatments provide prolonged relief from symptoms, slow down disease progression, and deliver optimal care for patients with SS.

This study aimed to examine the long-term effects of various treatment modalities, focusing particularly on time to next treatment (TTNT), while describing the demographic, clinical, histopathological, and molecular features of patients at diagnosis, along with their comorbidities and outcomes.

## 2. Materials and Methods

This was a descriptive, retrospective, register-based study of patients diagnosed with SS. A total of 171 patients (including deceased patients) with an initial diagnosis of PCL who had at some point been referred to the dermatology clinic at Sahlgrenska University Hospital were identified from the PCL register in West Sweden between 1 January 2005 and 1 March 2024. This register is hosted at the Department of Dermatology and Venereology, Sahlgrenska University Hospital, and was founded in 2014. Data on patients with any PCL are continuously added to this register. After a systematic review of medical records, histological analysis of skin biopsies, and blood involvement, 17 patients were identified as having SS; the remainder had other PCLs (Figure 1).

All patients who were not previously staged at the time of diagnosis were staged according to the current ISCL/WHO-EORTC classification based on information from medical records and photographic documentation from the time of diagnosis [1,8,13,21,22]. Patients were considered eligible if confirmation of B2 blood involvement was present, and the date of confirmation was set as the date of debut [8,12,13]. Demographic information was collected and analyzed along with clinical, histopathological, immunohistochemical, and flow cytometry findings. Information about comorbidities was collected from the patients’ medical records.

The treatment response was estimated based on data from medical records and photographic documentation from follow-up [12]. Response to treatments was evaluated in terms of the response criteria defined for SS by the ISCL/WHO-EORTC [12], with the addition of TTNT in line with the standardized definition proposed by Campbell et al. [23]. TTNT was measured from the start date of one course of treatment to the start date of the next, ignoring short-term treatment gaps that lasted <1 month. For patients who did not receive a subsequent line of treatment, TTNT was censored at the date of death or at last recorded follow-up. Addition of another systemic therapy was considered to constitute a new line of treatment, thus triggering a TTNT event for the existing therapy. Skin-directed therapies such as topical steroids, chlormethine gel, ultraviolet light A (UVA1), ultraviolet light B (UVB), psoralen plus ultraviolet light A (PUVA), and radiation therapy were excluded from TTNT evaluation. 

All data were analyzed using version 3.5.3 of R (The R Foundation for Statistical Computing, Vienna, Austria). Kaplan–Meier plots were generated for overall survival and were compared using log-rank tests. All tests were two-sided, and *p* < 0.05 was considered to be statistically significant. Microsoft Excel was used to create tables and graphs. Descriptive statistics were calculated for baseline characteristics such as age, Modified Severity-Weighted Assessment Tool (mSWAT) score, and duration of symptoms, either as medians with 95% confidence intervals (CIs) or as percentages. 

## 3. Results

### 3.1. Demographic and Clinical Characteristics

This cohort comprised 17 patients in West Sweden diagnosed from January 1, 2012 to March 1, 2024: 12 diagnosed with SS de novo, 3 with secondary SS (preceded by MF), and 2 with MF with leukemic blood (B2) involvement. All participants had Swedish ethnicity. The median age at diagnosis was 68 years (range: 54–86 years), with a male-to-female ratio of 1.8:1 (Table 1). Most of the patients (n = 13, 76.5%) were classified as stage IVA1, while three (17.7%) were categorized as IVA2 and one (5.9%) as IVB [1,8,13,21,22] (Table 1).

The range of occupations was diverse, encompassing both manual labor (house painter, truck driver) and more sedentary or managerial positions (accounting consultant and project manager). Additionally, some patients had multiple occupations or worked in varied settings such as farms or schools (Appendix A). Photographs of the patients are presented in Figure 2, while details of their clinical characteristics can be found in Table 1 and Appendix A.

At the time of diagnosis, 12 patients (71%) had erythroderma, 8 (47%) had clinically confirmed lymphadenopathy, and all 17 (100%) showed leukemic (B2) involvement with a Sézary cell count of ≥1000/μL in peripheral blood (Table 1 and Table 2).

### 3.2. Comorbidities

Throughout the entire course of the disease, the most common comorbidities in our study population were cardiovascular diseases, including hypertension (82%), hyperlipidemia (41%), diabetes mellitus (29%), and congestive heart failure (29%). Additionally, four patients (24%) had benign prostatic hyperplasia, osteoarthritis, and cataracts; three (18%) were diagnosed with pulmonary embolism, deep vein thrombosis, and atrial fibrillation (Appendix A); and eight had other malignancies in addition to SS (Table 2). All patients had pruritus.

Around a third of the patients (35%) were classified as obese (body mass index [BMI] ≥ 30 kg/m^2^), another third (30%) were classified as overweight (BMI 25.0–29.9 kg/m^2^), and the remaining third (35%) were classified as healthy weight (BMI 18.5–24.9 kg/m^2^). Among the 16 patients for whom smoking information was available, 8 (50%) were former smokers and the remaining 8 (50%) had never smoked; none were current smokers.

### 3.3. Histopathological, Immunohistochemical, and Flow Cytometry Findings

Histopathological analysis of skin biopsies was performed in eight patients (50%) upon diagnosis, in six patients (37.5%) during follow-up, and in two patients (12.5%) immediately before the diagnosis; the remaining one patient was not biopsied (Table 3). Six patients underwent histologically confirmed large-cell transformation, characterized by the transition from small-to-intermediate-sized cells to large cells [24] throughout the disease. A summary of the immunohistochemistry and flow cytometry results is given in Table 3.

### 3.4. Treatment Modalities

Treatments were categorized into 15 systemic monotherapies (51 treatments), 12 systemic combination therapies (36 treatments), and 6 non-systemic treatments (23 treatments). It was common for the same therapy to be re-administered. Figure 3 shows all treatments received by the 17 patients between March 2012 and March 2024. An overview of TTNT analysis for systemic monotherapies and combination therapies is given in Table 4, and treatment outcomes for systemic monotherapies and skin-directed therapies are presented in Table 5. The median duration of all treatments was 5.88 months (95% CI: 4.17–6.72).

#### 3.4.1. Treatment Groups

Among the three different treatment groups, triple therapy involving retinoids (alitretinoin or bexarotene), interferon alpha (IFN-α), and extracorporeal photopheresis (ECP) demonstrated the longest TTNT, at 14.14 months (95% CI: 2.52–24.05). In contrast, double therapy had a TTNT of 5.14 months (95% CI: 3.16–6.98) and monotherapy had a TTNT of 5.88 months (95% CI: 3.78–6.77) (*p* = 0.16; Table 4 and Figure 4).

#### 3.4.2. Lines of Treatment

The median number of systemic therapy lines per patient was 4 (range: 1–15). We found no significant difference in the TTNT among various lines of treatments. The median TTNT was 5.88 months (95% CI: 2.2–8.6) for first-line treatment, 7 months (95% CI: 4.9–12.9) for mid-line treatments (2–4 treatments), and 4.2 months (95% CI: 2.9–6.5) for late-line treatment (≥5 treatments) (*p* = 0.146).

#### 3.4.3. Systemic Monotherapies

Monotherapy was used throughout the entire course of the disease, most commonly (88% of patients) as first-line therapy. Acitretin was the most frequently used monotherapy (nine times), followed by bexarotene (eight times), ECP (seven times), and methotrexate (six times).

Allogeneic stem cell transplantation (allo-SCT) had the longest TTNT (39.7 months), followed by interferon-α (INF-α) (15.5 months) and isotretinoin (12.6 months) (Table 4). However, the extended duration of TTNT for isotretinoin was based on a single patient, who discontinued treatment due to adverse effects and chose not to pursue further therapy despite recommendations. 

Complete response was observed in 14.3% of cases treated with ECP and 50% of cases treated with CHOP. Additionally, PR was noted in 100% of cases treated with INF-α, alemtuzumab, doxorubicin, gemcitabine, and allo-SCT. ECP demonstrated a 71.4% PR rate (Table 5).

#### 3.4.4. Systemic Combination Therapies

We found that 59% of the patients received combination therapy, with the most prevalent combination being ECP and retinoids (used 13 times); followed by triple therapy consisting of ECP, IFN-α, and retinoids (used 8 times); and then IFN-α and retinoids (used 7 times). The longest median TTNT was observed in triple-therapy ECP with IFN-α and retinoid (14.1 months), followed by ECP with methotrexate (12.4 months) and ECP with retinoids (6.4 months) (Table 4).

#### 3.4.5. Non-Systemic Therapies 

The most frequently used skin-directed therapies were UVB (used 12 times) and PUVA (used 5 times). Topical corticosteroids were not considered as a separate treatment course in this study. PR rates were most favorable in PUVA therapy (80%) followed by UVA1 therapy (66.7%), while UVB therapy showed a substantially lower PR rate of only 33.3% (Table 5).

#### 3.4.6. First-Line Treatments

The most frequently used first-line treatments (in 65% of cases) were retinoid monotherapy (acitretin: 35%; bexarotene: 29%). ECP both as monotherapy and as combination therapy was less commonly used, comprising only 24% of cases.

There was a significant difference in median TTNT between patients who received ECP as first-line therapy (3.48 months) and those who received other treatments (6.37 months) as first-line therapy (*p* = 0.027) (Figure 5).

#### 3.4.7. ECP Treatment 

ECP was used in 53% of the patients, and 78% of these received ECP within one year of diagnosis. TTNT was longer for ECP combination therapy than for monotherapy (6.36 vs. 1.44 months; *p* = 0.31; Table 4), and significantly longer for patients receiving ECP as late-line treatment than for those receiving first-line ECP (6.41 vs. 3.48 months; *p* = 0.0043). No statistically significant differences were observed between patients receiving all ECP treatments and those undergoing all other non-ECP treatments (*p* = 0.46).

### 3.5. Disease Progression

Table 6 presents clinical outcomes and detailed information for 17 patients with SS. Among the seven deceased patients, four deaths (57%; cases 11, 12, 15, and 16) were associated with Sézary syndrome, while the remaining three (43%) were attributed to other causes including cardiac arrest, cardiorenal syndrome, and sepsis (Table 1). The median time to death was 2.14 years (range: 0.4–5.9 years). Overall survival rates declined over time, from 94% (95% CI: 84–100%) at 1 year, with a slight decrease to 88% (95% CI: 73–100%) at 2 years, and a more substantial drop to only 60% (95% CI: 39–92%) at 5 years (Figure 6).

Median survival showed notable differences across different stages: 40.8 months (95% CI: 24.1–86.7) for all stages combined; 40.8 months (95% CI: 24.2–86.7) for stage IVA1; 71.3 months (95% CI: 20.9–95.4) for stage IVA2; and only 5.2 months for stage IVB. Patients with MF+B2 had the longest median survival, at 50.7 months (95% CI: 10.2–91.2), followed by those with secondary SS, at 42.8 months (95% CI: 24.1–108.6). Those with de novo SS had the shortest median survival, at 35.2 months (95% CI: 24.9–79).

## 4. Discussion

### 4.1. General Results 

This retrospective study of 17 SS patients showed a median survival of only 2.1 years, underscoring the urgent need for effective diagnosis and treatments to improve outcomes in SS. We previously published a similar study exclusively concentrating on patients with MF in West Sweden [25]. Now, we aim to distinguish between MF and SS as two distinct diseases, and have focused on patients with B2 blood involvement to gain a more profound understanding of the disease.

### 4.2. Clinical Characteristics

At the time of diagnosis, only 35% of patients showed the classical triad of symptoms, which suggests that these symptoms are not mandatory for a diagnosis. A prior study by Kamijo et al. reviewed 37 cases of SS without initial erythroderma, 12 of which exhibited only lymphadenopathy [26]. Five patients in our study did not initially present with erythroderma. Of these, three developed it later, while the other two were diagnosed with MF at stage B2 due to erythroderma absence during the course. One of the MF patients displayed papules, patches, plaques, and onychodystrophy, while the other presented plaques, nodules, and keratoderma. It is essential to note whether patients exhibit keratoderma (74.5%), ectropion (64.7%), or onychodystrophy (64.7%), as many of our patients presented with these clinical features (Table 1 and Appendix A).

Eight of the ten patients with *S. aureus* colonization were treated with antibiotics, and four showed a PR with clinical improvement; the four who showed no improvement all had disease progression (Table 1). We found no significant correlation between *S. aureus* colonization and the progress of the disease, but this could be due to the small number of patients. More research on this topic should be conducted in the future.

### 4.3. Demographic Factors

The majority (70.6%) of the patients were engaged in physical occupations with varying levels of physical activity, ranging from moderate to heavy work (Appendix A). A previous study found that farmers, painters, and carpenters were at higher risk of MF/SS [27], and 24% of the patients in the present study had this type of occupation. Other studies have also reported exposure to chemicals as being associated with advanced-stage CTCL disease [28,29]. We can therefore assume that patients working in professions known to have more exposure to substances than others, such as firefighters, house painters, or farmers, could have an increased risk of developing SS.

### 4.4. Comorbidities

All 17 patients had comorbidities other than SS. Three patients had experienced ischemic stroke, and two had suffered myocardial infarctions. A Danish cohort study of 483 patients with CTCLs showed a higher risk of stroke or heart attack within 5 years of diagnosis [30]. In contrast, two retrospective studies of CTCL patients conducted in Finland [31] and the USA [32] found no increased risk of coronary artery disease. We observed high prevalences of hypertension (82%), type 2 diabetes mellitus (29%), and hyperlipidemia (41%) (Appendix A). 

Similarly to our previously published study on MF, almost half (47%) of our patients with SS had other malignant comorbidities [25] (Table 2). Four (24%) had secondary skin malignancies. A previous study by Scheu et al. suggested that increased exposure to UV radiation can be a factor in whether a patient develops secondary malignant skin tumors [33]. In the present study, only 3 (27%) of the 11 patients who underwent phototherapy (UVA, UVB, PUVA) developed skin malignancies. Further investigation of this association should be conducted in larger cohorts.

### 4.5. Diagnostic Findings

#### 4.5.1. Histopathological Findings

The histopathology was non-diagnostic in nearly 20% of our cases, while flow cytometry provided a diagnostic outcome in 100% of cases. Previous studies have reported similar conclusions, indicating that histopathology in SS could be non-diagnostic [34,35]. Another contributing factor could be that in 37.5% of cases (n = 16), a skin biopsy was performed during the disease, potentially leading to diagnostic challenges as patients were already undergoing treatment. A total of six patients had undergone exclusively skin-directed therapy before the skin biopsy. The interesting aspect here is that the skin biopsies could show similar results regardless of whether they were obtained at the first diagnosis or during follow-up, as previously seen in a study of 57 SS cases [36]. 

#### 4.5.2. Immunohistochemical Findings

As previously described, nearly all SS patients in our cohort exhibited clonal T-cells CD3+ (86.7%) and CD4+ (100%); however, our finding that only 20% had CD8− was in contrast to previous results [37,38,39,40]. We also noted a low count of CD8+ lymphocytes, which can indicate a diagnosis of SS [36].

#### 4.5.3. Flow Cytometry Findings

We found that 87.5% of patients had increased levels of CD4+CD26− and 58.3% had increased levels of CD4+CD7− in peripheral blood at the time of diagnosis (Table 3). This result is consistent with an Italian retrospective study, where over 90% of 107 SS patients demonstrated aberrant CD26 expression; however, CD7 expression varied between patients [41]. 

Another study showed that flow cytometry findings are highly specific (100%) and sensitive (>80%) for patients with SS and loss of CD7 (≥40%) and/or CD26 (≥80%) [42]. There are only a few studies including MF and SS together that have investigated the relationship between flow cytometry findings, treatment outcomes, and disease progression [41]. The present study focused only on SS, but we found no significant relationship between flow cytometry findings and disease progression with death due to lymphoma (Table 6).

#### 4.5.4. Molecular Findings

Genetic profiling can provide valuable insights into the underlying mechanisms of SS; however, we did not conduct a molecular study for genetic profiling of SS cases.

### 4.6. Treatment Regimens and Response

The initial management of SS patients varies, as there are several treatment options [14,42]. The median duration of all treatments among our patients was 5.88 months, which is consistent with previous reports [14,43]. 

In our study, retinoids were primarily used as first-line therapy, comprising 65% of cases, and were commonly employed across all treatment regimens, either alone (45%) or in combination with ECP (58%). TTNT for all retinoids was 5.85 months, which was higher than the 4.4 months reported in another study [14]. When retinoids were used in combination therapy with ECP, the TTNT was 6.38 months, and in combination with IFN-α, this extended to 14.14 months. A previous review found that the combination of ECP, IFN-α, and retinoids (bexarotene) potentially yielded the highest response rates in patients with SS [44]. In line with this, when used among our patients, the triple therapy regimen comprising retinoids (alitretinoin or bexarotene), IFN-α, and ECP demonstrated the longest median TTNT, at 14.14 months, compared to both double therapies and monotherapies (5.16 vs. 5.88 months) (Table 4). 

Combination treatment with ECP, IFN-α, and retinoids provided 1 year free from further treatment in 50% of cases and 2 years free from treatment in 25% of cases. These durations exceed those found in a previous study concentrating on ECP-based combination therapy, which reported rates of 40.1% at 1 year and 14.9% at 2 years [14]. 

When examining individual outcomes, it is important to emphasize that allo-SCT is the only potentially curative option [14,45]. In our study, this treatment demonstrated the longest TTNT (39.72 months) but was only used in one patient. 

Two patients treated with IFN-α achieved a TTNT of 15.51 months, with a predicted 50% (95% CI: 1.3–98.7) being free from the next line of treatment at 1 year; these are substantially better figures than seen in previous reports of a 4.8 month TTNT and 14.3% 1-year freedom from treatment [14].

In our cohort, therapy based on monoclonal antibodies accounted for only 11.8% of monotherapy and 5.6% of combination therapy. This is likely due to restrictive prescribing policies.

PUVA exhibited the highest PR rate at 80%, followed by UVA1 at 66.7%, and UVB had a lower response rate of 33.3%; however, due to the predominance of light skin types among patients (71% had skin type I or II), UVB was preferred over PUVA.

#### Treatment with Extracorporeal Photopheresis

ECP monotherapy demonstrated an overall response rate of 85.7%, which is consistent with previous findings of 63% (43–100%), while both ECP combination therapy and ECP monotherapy had lower TTNT (6.36 and 1.44 months, respectively) compared to previous results (9.2–12 months) [14,43,46]. Earlier research has emphasized the significance of ECP as the primary treatment for SS patients [14,47]. However, our results suggest that patients initially treated with non-ECP-containing therapy experienced a significantly longer TTNT compared to those treated with ECP-containing therapy (ECP monotherapy plus ECP combination therapy), with a median TTNT of 6.36 versus 3.48 months (*p* = 0.027). We observed that administering ECP earlier did not result in improved outcomes, as patients who received ECP as a late-line treatment experienced a significantly longer TTNT than those who received it as a first-line treatment (6.41 vs. 3.48 months; *p* = 0.0043). Following the latest EORTC consensus recommendations for SS treatment, ECP may be integrated into combination therapy. However, there is limited evidence suggesting the superiority of one combination therapy over another [42,47]. 

More than half of the patients in this study had a combination of treatments, and 80.6% of these combinations involved ECP. We found that the effect of ECP may increase when combining it with other therapies, as seen in some previous [48,49] studies, since the TTNT was 1.48 months with ECP monotherapy versus 6.37 months with ECP combination therapy; however, this difference was not significant (*p* = 0.31). We observed that 53% of our patients received ECP treatment, with 78% receiving it within 1 year of diagnosis. However, it is important to clarify that most of the differences in ECP treatment seen between this study and previous studies may be attributed to the limited number of patients.

Previous studies have reported that the median survival with ECP treatment is 6–8 years, compared to 29–62 months for SS patients without ECP treatment [49]. Our findings were similar, with a median survival of 5.9 years (range: 0.43–9.41 years) among ECP-treated patients and 24.9 months (10.2–91.2) among those without ECP treatment (*p* = 0.61). However, as noted by Alfred et al., patients with lower tumor burden respond better to ECP treatment and are more frequently selected for this treatment, which might have led to bias in the results both in previous studies and in our study [49].

### 4.7. Staging and Disease Progression 

In comparison to a previous systematic review and meta-analysis, which reported median survival times of 23–53 months for stage IVA1, 13–29 months for stage IVA2, and 1.4 months for stage IVB, our study showed longer median survival for stages IVA2 (71.3 months) and IVB (5.2 months), but similar survival for stage IVA1 (40.8 months) [50]. 

Three patients with histological involvement of lymph nodes in stage IVA2 also had de novo SS, but no firm conclusions can be drawn from this due to the limited number of patients. Although patients with de novo SS showed lower median survival than those with secondary SS or MF+B2, the difference was not statistically significant (*p* = 0.59). An earlier study demonstrated better outcomes for patients with de novo advanced-stage disease (IIB-IVB including SS), though as with our findings, the difference was not statistically significant (*p* = 0.25) [51].

### 4.8. Limitations

This study has several limitations. Some of the patients had only a short observation period, the sample size was small, and the data were obtained retrospectively, which limited the power of our study. Despite these limitations, we believe that our analysis of these 17 individuals provides valuable insights. However, our primary focus remains on delivering essential information to address the urgent need for deeper understanding of SS and enhancing the care of SS patients. This was a single-center study, and so the results may not fully reflect the situation in other regions due to variations in the availability of treatments, regional regulatory prescribing restrictions, and individual physician practice. Although national guidelines do exist, there is currently no national register in Sweden. If such a register was set up, then future studies would be able to utilize data from the entire country. Another limitation arising from the small sample size is that a single patient refusing or not receiving treatment for some time might have had a large impact on the results.

The therapeutic possibilities for SS are continuously expanding. However, due to access and prescribing restrictions, we were not able to use all of the treatments outlined in the latest guidelines from 2023 [42] or treatments with histone deacetylase inhibitors (HDACi) that have shown promise in the treatment of SS [52]. Most of our patients had already started their treatments by the time these guidelines were published.

## 5. Conclusions

This retrospective study examined demographic factors, clinical features at diagnosis, and treatment modalities among 17 SS patients in West Sweden. The findings highlight the importance of personalized diagnostic approaches, as not all patients presented with the classical triad of symptoms at diagnosis. Combination therapy, particularly a triple regimen of retinoids, IFN-α, and ECP, emerged as a promising first-line treatment. 

This study emphasizes the necessity of enhanced treatment standards for SS patients in Sweden, comparable to global standards, and underscores the importance of future prospective studies. Effective diagnostic and therapeutic strategies for rare diseases such as SS can improve patient care worldwide.

## Figures and Tables

**Figure 1 cancers-16-01948-f001:**
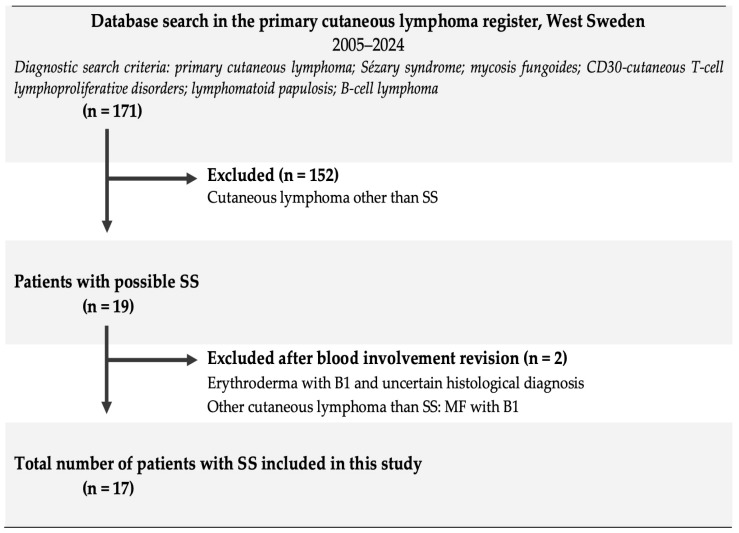
Flowchart showing the applied inclusion and exclusion criteria. B1: low blood involvement (>500 and <1000 cells/μL); CD: cluster of differentiation; MF: mycosis fungoides; SS: Sézary syndrome.

**Figure 2 cancers-16-01948-f002:**
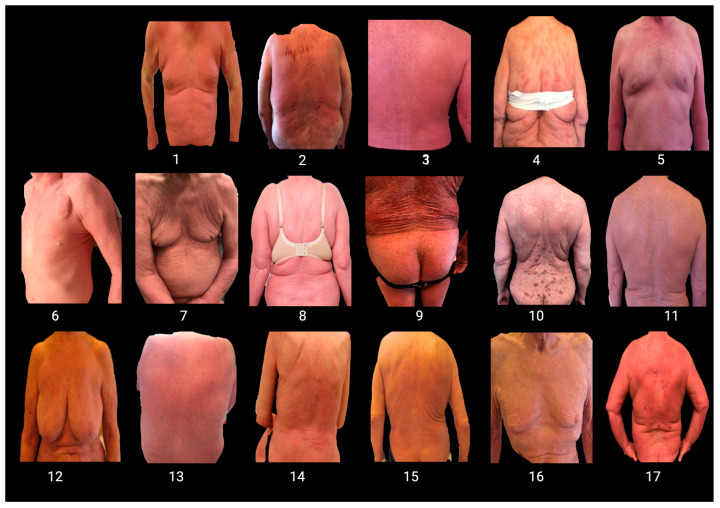
Photographs representing 17 unique patients diagnosed with Sézary syndrome.

**Figure 3 cancers-16-01948-f003:**
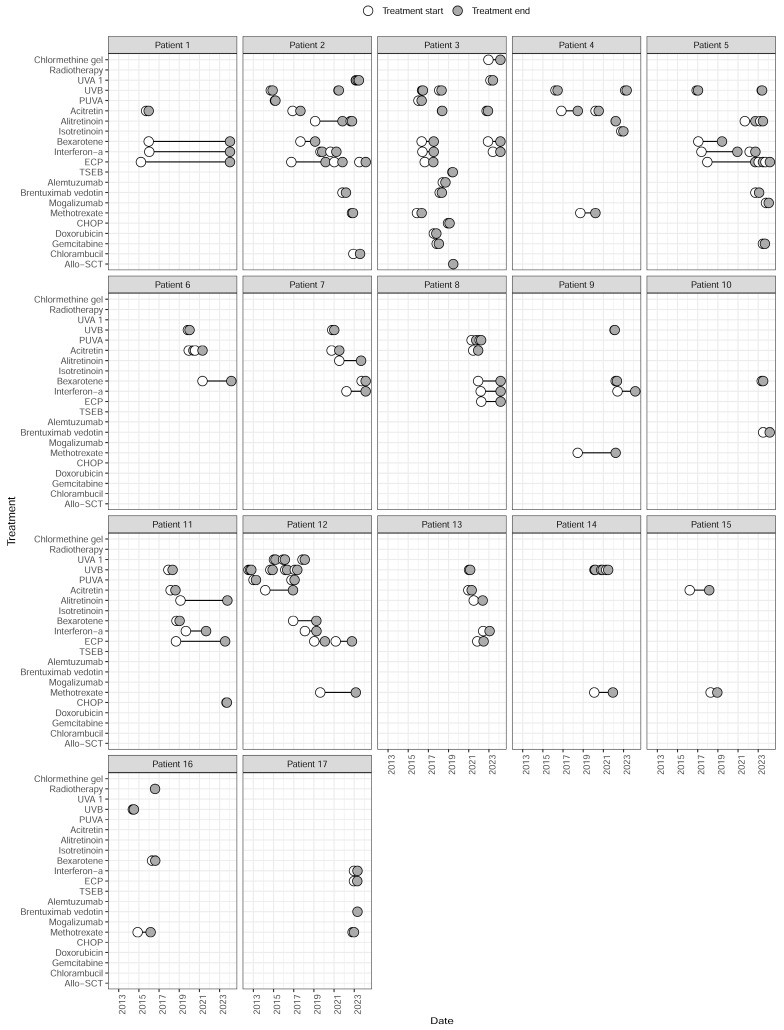
Treatments received by 17 patients with Sézary syndrome between 2012 and 2024, including former and current regimens after B2 involvement in blood. Allo-SCT: allogenic stem cell transplantation; CHOP: cyclophosphamide, doxorubicin hydrochloride, Oncovin, prednisone; ECP: extracorporeal photopheresis; PUVA: psoralen plus ultraviolet light A; TSEB: total skin electron beam; UVA1: ultraviolet light A; UVB: ultraviolet light B.

**Figure 4 cancers-16-01948-f004:**
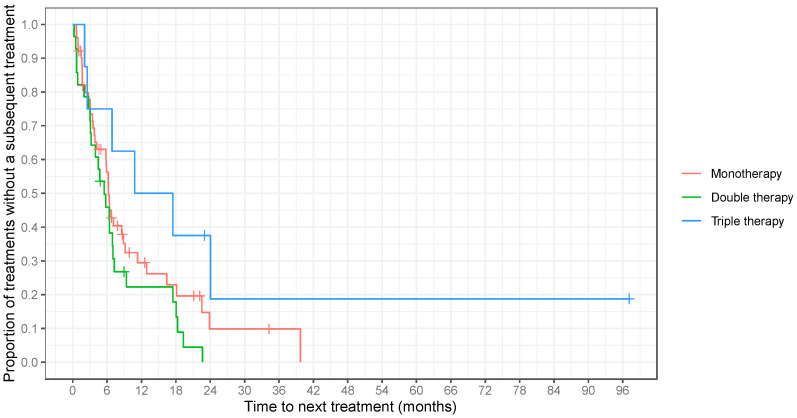
Time to next treatment (TTNT) did not differ significantly between monotherapy, double therapy, and triple therapy (median TTNT: 5.9, 5.1, and 14.1 months, respectively; *p* = 0.075). TTNT was measured from the initiation date of one treatment regimen to the start date of the subsequent course of treatment. Vertical lines denote censoring at death or last follow-up.

**Figure 5 cancers-16-01948-f005:**
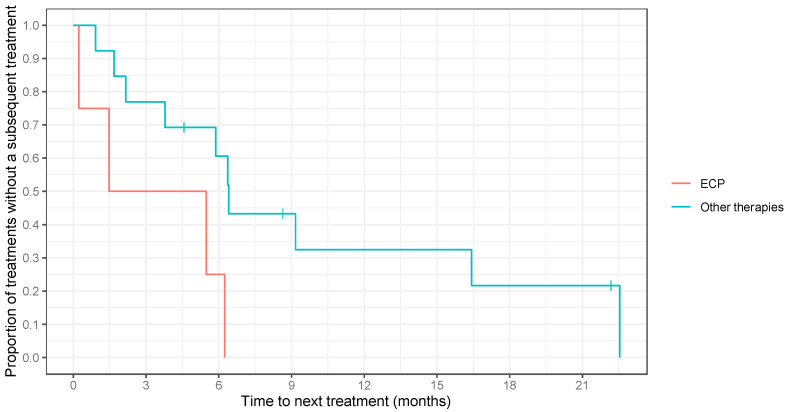
Extracorporeal photopheresis (ECP) as first-line therapy versus all other treatments as first-line therapy (*p* = 0.027). Time to the next treatment was measured from the initiation date of one treatment regimen to the start date of the subsequent course of treatment. Vertical lines denote censoring due to death.

**Figure 6 cancers-16-01948-f006:**
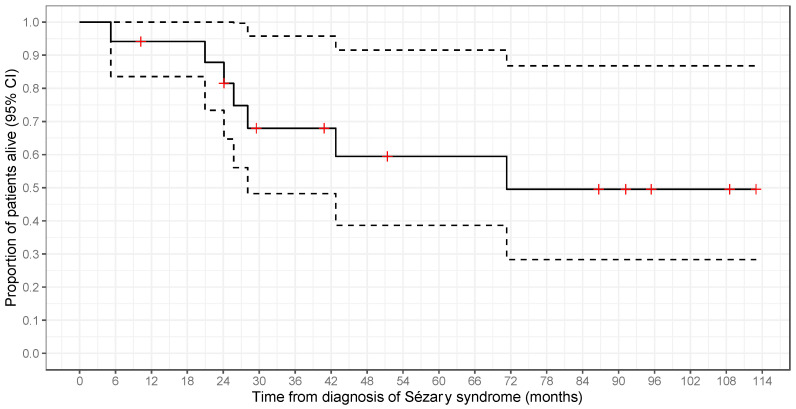
Kaplan–Meier survival plot illustrating overall survival after the diagnosis of Sézary syndrome. The dotted lines represent the 95% confidence intervals (CI). Vertical lines indicate patients lost to follow-up.

**Table 1 cancers-16-01948-t001:** Demographic and clinical features of 17 patients with Sézary syndrome.

Characteristics	Values
Age in years at time of diagnosis, median (range)	68.2 (54–85.6)
Mean ± SD	70.7 ± 9.3
Sex, n (%)	
Male	11 (64.7)
Fitzpatrick skin type, n (%)	
I	2 (11.8)
II	10 (58.8)
III	5 (29.4)
Duration in years of skin symptoms before initial diagnosis, median (range)	1.07 (0.24–10.62)
Clinical features during the disease, n (%)	
Onychodystrophy	11 (64.7)
Ectropion	11 (64.7)
Leonine facies	2 (11.8)
Alopecia	4 (23.5)
Keratoderma	13 (74.5)
Erythroderma	15 (88.2)
Erythroderma at debut	12 (70.6)
mSWAT score at time of diagnosis, median (range)	66 (9–106)
Mean ± SD	57.7 ± 30.3
Clinical stage at time of diagnosis, n (%)	
IVA1	13 (76.5)
IVA2	3 (17.7)
IVB	1 (5.9)
Types, n (%)	
De novo SS	12 (70.6)
Secondary SS	3 (17.6)
MF with B2	2 (11.8)
Skin colonization with *Staphylococcus aureus* at any time, n (%)	10 (58.8)
Treatment of *Staphylococcus aureus*, n (%)	7 (41.2)
Increased LDH level at time of diagnosis, n (%)	5 (41.7)
Increased LDH level during the disease, n (%)	12 (70.6)
Median LDH level, units/liter (IQR)	4.5 (3.9–5.9)
Increased eosinophil level at time of diagnosis, n (%)	0
Increased eosinophil level during the disease, n (%)	6 (42.9)
Eosinophil level in cells/μL, median (IQR)	0.6 (0.48–0.85)
Time in years from diagnosis to last observation, median (range)	3.4 (0.43–9.4)
Deceased patients, n (%)	7 (41.2)
SS-related death	4 (57.1)
Other causes of death	3 (42.9)

IQR: interquartile range; LDH: lactate dehydrogenase; MF: mycosis fungoides; mSWAT: Modified Severity-Weighted Assessment Tool; SD: standard deviation; SS: Sézary syndrome.

**Table 2 cancers-16-01948-t002:** Malignant comorbidities in eight patients with Sézary syndrome.

	Total	Before Sézary Syndrome Diagnosis	After Sézary Syndrome Diagnosis
Skin cancers			
Basal cell carcinoma	3	1	2
Melanoma in situ	1		1
Atypical fibroxanthoma	1		1
Hematological malignancies			
Diffuse large B-cell lymphoma	1		1
Mantle cells lymphoma	1	1	
Solid tumors			
Prostate cancer	1		1
Lung cancer	1		1
Rectal/colon cancer	2	1	1

**Table 3 cancers-16-01948-t003:** Diagnostic criteria in 17 patients with Sézary syndrome.

Characteristics	Frequency, n (%)
Histopathological characteristics, skin	
*Epidermotropism*	
Yes	12 (75)
No	3 (18.8)
Uncertain	1 (6.25)
*Atypical lymphocytes*	
Yes	13 (81.25)
No	3 (18.8)
TCR clonality, skin	
Monoclonal	15 (100)
Gamma	15 (100)
Beta	13 (86.7)
Immunohistochemistry, skin	
CD2+	3 (20)
CD3+	13 (86.7)
CD4+	15 (100)
CD5+	3 (20)
CD7−	2 (13.3)
CD8−	3 (20)
CD8+	3 (20)
CD20+	1 (6.7)
CD30+	3 (20)
Lymphadenopathy (clinically)	8 (47.1)
Lymph node involved (histopathologically)	5 (29.4)
Bone marrow biopsy performed	8 (47.1)
Bone marrow involved	5 (62.5)
Blood characteristics	
CD4/CD8 ratio ≥ 10	11 (64.7)
CD4+CD7− ≥ 40%	7 (58.3)
CD4+CD26− ≥ 30%	14 (87.5)
Sézary cell count in cells/µL, median (IQR)	1428 (1206–1976)

CD: cluster of differentiation; IQR: interquartile range; TCR: T-cell receptor.

**Table 4 cancers-16-01948-t004:** TTNT analysis for systemic monotherapies and systemic combination therapies.

	n	TTNT (Months)		1 Year Free from Further Treatment, % (95% CI)	2 Years Free from Further Treatment, % (95% CI)	First Line of Therapy, n (%)	Lines of Therapy	
		Median	95% CI				n	Range
**Systemic monotherapies**	51	5.88	3.78–6.74	19.6 (9.8–33.1)	3.9 (0.5–13.5)	15 (88.24)	3	1–15
ECP	7	1.48	0.66–6.77			2 (11.76)	5	1–14
IFN-α	2	15.51	9.89–21.13	50 (1.3–98.7)			3.5	2–5
Methotrexate	6	10.78	3.3–20.16	50 (11.8–88.2)		2 (11.76)	2	1–8
Monoclonal antibody	6	5.04	1.89–9.56				7	2–9.5
Alemtuzumab	1	6.21					8	
Brentuximab-vedotin	5	3.88	1.05–11.30				6	2–11
*Multi-agent chemotherapy*	2	3.98	1.74–6.21				8.5	8–9
CHOP	2	3.98	1.74–6.21				8.5	8–9
*Retinoids*	23	5.85	3.78–6.74	22 (7.5–43.7)			2	1–4
Acitretin	9	6.41	5.88–22.54	33.3 (7.5–70.1)		6 (35.29)	1	1–7
Alitretinoin	5	5.85	1.61–8.54				4	2–7
Isotretinoin	1	12.58		100 (2.5–100)			5	
Bexarotene	8	3.40	1.92–.78	12.5 (0.3–52.7)	12.5 (0.3–52.7)	5 (29.41)	1	1–2
*Single-agent chemotherapy*	4	3.50	3.02–7.10				8	4–13
Doxorubicin	1	3.45					4	
Gemcitabine	2	3.29	3.02–3.55				8	5–11
Chlorambucil	1	7.10					13	
Allo-SCT	1	39.72		100 (2.5–100)	100 (2.5–100)		10	
**Systemic combination therapies**	36	6.06	3.6–8.11	25 (12.1–42.2)	5.5 (0.7–18.7)	2 (11.76)	4	1–14
*ECP based*	29	6.37	3.22–9.36	27.6 (12.7–47.2)			4	3–6
ECP + MTX	2	12.40	5.49–19.32	50 (1.3–98.7)		1 (5.88)	2	1–3
ECP + single-agent chemotherapy	1	0.85					14	
ECP + IFN-α	3	0.53	0.23–18.30	33.3 (0.8–90.6)		1 (5.88)	4	1–4
ECP + monoclonal antibody	2	2.6	1.97–3.22				11	9–13
ECP + retinoid	13	6.37	3.97–7.26	15.4 (1.9–45.4)			4	3–6
ECP + IFN-α + retinoid	8	14.14	2.53–24.05	50 (15.7–84.2)	25 (3.2–65.1)		4.5	3.5–7
*IFN-α based*	7	4.8	2.76–8.97	14.3 (0.36–57.8)			3	2–8
IFN-α + retinoid	7	4.8	2.76–8.98	14.3 (0.36–57.8)			3	2–8

CHOP: cyclophosphamide, doxorubicin hydrochloride, Oncovin, prednisone; CI: confidence interval; ECP: extracorporeal photopheresis; INF-α: interferon-α; MTX: methotrexate; TTNT: time to next treatment.

**Table 5 cancers-16-01948-t005:** Treatment outcomes for systemic monotherapies and skin-directed therapies in 17 patients with Sézary syndrome.

	n	CR, n (%)	PR, n (%)	SD, n (%)	PD, n (%)	NA, n (%)
**Systemic monotherapies**	51					
Extracorporeal photopheresis	7	1 (14.3)	5 (71.4)		1 (14.3)	
Interferon-α	2		2 (100)			
Methotrexate	6			3 (50)	2 (33.3)	1 (16.7)
Monoclonal antibody-based therapy	6					
Alemtuzumab	1		1 (100)			
Brentuximab-vedotin	5		3 (60)	1 (20)	1 (20)	
Multi-agent chemotherapy	2					
CHOP	2	1 (50)			1 (50)	
Retinoids	23					
Acitretin	9		3 (33.3)	3 (33.3)	3 (33.3)	
Alitretinoin	5		2 (40)	2 (40)	1 (20)	
Isotretinoin	1			1 (100)		
Bexarotene	8		2 (25)	4 (50)	2 (25)	
Single-agent chemotherapy	4					
Doxorubicin	1		1 (100)			
Gemcitabine	2		2 (100)			
Chlorambucil	1			1 (100)		
Stem cell transplantation allogenic	1		1 (100)			
**Skin-directed therapy**	23					
Ultraviolet light A1	3		2 (66.7)	1 (33.3)		
Ultraviolet light B	12		4 (33.3)	4 (33.3)	2 (16.7)	2 (16.7)
Psoralen and ultraviolet light A	5		4 (80)		1 (20)	
Radiotherapy	1				1 (100)	
Total skin electron therapy	1	1 (100)				
Chlormethine gel	1		1 (100)			

CHOP: cyclophosphamide, doxorubicin hydrochloride, Oncovin, prednisone; CR: complete response; NA: not applicable; PD: progressive disease; PR: partial response; SD: stable disease.

**Table 6 cancers-16-01948-t006:** Age, stage, type, blood data, clinical outcome, and information about combination therapy for 17 patients with Sézary syndrome (SS). The cases are presented in descending order based on follow-up time, regardless of clinical outcome.

Case	Age ^a^	Stage ^b^	Stage ^c^	Type	Sézary Cells ^d^	Cell Ratios at Diagnosis			Transformation	Disease Progression	Follow-Up Time ^e^	Clinical Outcome	Combination Therapy
						CD4/CD8	CD4+/CD7−	CD4+/CD26−					
1	56.8	IVA1	Remission	De novo SS	1045	5	NA	84	No	No	9.4	Alive	No
2	60.1	IVA1	II	Secondary SS	1428	14	NA	76	Yes	Yes	9.0	Alive	No
3	54.0	IVA2	IA	De novo SS	1078	7	NA	77	Yes	Yes	8.0	Alive	Yes
4	79.2	IVA1	IVA1	MF with B2	2046	20	56	76	No	No	7.6	Alive	Yes
5	66.1	IVA1	IIIA	De novo SS	3000	256	3	92	No	Yes	7.2	Alive	No
6	79.7	IVA1	IVB	De novo SS	1845	2	NA	62	No	No	4.3	Alive	Yes
7	85.6	IVA1	IIIB	De novo SS	1620	20	43	45	No	No	3.4	Alive	Yes
8	66.4	IVA1	IVB	De novo SS	6786	20	78	78	No	Yes	2.5	Alive	Yes
9	80.0	IVA1	IA	De novo SS	2028	38	17	21	No	No	1.9	Alive	No
10	68.0	IVA1	IIIB	MF with B2	1206	4	38	42	Yes	Yes	0.9	Alive	Yes
11	70.3	IVA2	IA	De novo SS	1288	18	59	60	No	Yes	5.9	Deceased	Yes
12	78.7	IVA1	IB	Secondary SS	1314	21	23	73	NA	Yes	3.6	Deceased	No
13	63.8	IVA1	IVA1	De novo SS	1155	10	75	75	No	Yes	2.3	Deceased	Yes
14	68.2	IVA1	IVA1	De novo SS	1976	5	17	33	No	No	2.1	Deceased	No
15	80.9	IVA1	IVA1	Secondary SS	1292	16	NA	NA	Yes	Yes	2.0	Deceased	No
16	77.4	IVA2	IVA2	De novo SS	1551	1.5	47	22	Yes	Yes	1.7	Deceased	Yes
17	66.6	IVB	IVB	De novo SS	1012	20	54	78	Yes	No	0.4	Deceased	Yes

a: age at diagnosis of SS; b: stage at diagnosis of SS; c: last recorded stage; d: median Sézary cell count in cells/µL at diagnosis; e: time in years from diagnosis to the last recorded follow-up; CD: cluster of differentiation; MF: mycosis fungoides.

## Data Availability

The data presented in this study are available upon reasonable request to the corresponding author.

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
