# Peer review of "Sézary Syndrome in West Sweden: Exploring Epidemiology, Clinical Features, and Treatment Patterns in a Registry-Based Retrospective Analysis"

_cancers, 2024, doi:10.3390/cancers16111948_

Round 1
Reviewer 1 Report
Comments and Suggestions for Authors
In the present study, Wojewoda et al. retrospectively analyzed epidemiologic, clinical and therapeutic data of 17 Sézary Syndrome (SS) patients from a single center in Sweden between 2012 and 2024. It is true that the sample size is low, but the study is important, presenting data regarding SS, the rare and aggressive form of CTCL. Such studies deserve to be published in order to better understand these rare forms of cancers. The manuscript is well written and the results are clear.
Please find below some comments that need to be addressed:
- Weirdly, I did not see any SS patient treated with HDACi (i.e. Vorinostat, Romidepsin) in this study. Is there a specific reason behind that? Or any national regulation (maybe it is not the case, as stated in the limitation section)? Knowing that these drugs are approved for the treatment of SS. And several centers use these drugs for refractory or relapse CTCL.
I guess this idea should be discussed and clearly stated within the manuscript (PMID: 17962618, 16960145, 33229141, and https://doi.org/10.1016/j.fander.2023.03.006 in french). And in both cases, a section talking about HDACi should be added.
Also, these drugs are reported to impact many aspects of cancer cells in CTCL as single agents or in combination (PMID 35408897, 34765562, 28695331, 30529073).
-Also, no genetic data available in the manuscript. Is there any molecular testing performed for these patients? NGS panels? If yes, please also include this information. Or explain. Especially that the detection of genetic alterations is known to impact drug response.
-Minor comment: Table 6 is not correctly placed in the text.
-Maybe another limitation: absence of data regarding the use of HDACi
Author Response
Cancers (ISSN 2072-6694)
AUTHORS COMMENTS TO THE EDITOR AND REVIEWER
Manuscript ID cancers: 3015464
Wojewoda et al
Title: Sézary Syndrome in West Sweden: Exploring Epidemiology, Clinical
Features, and Treatment Patterns in a Registry-Based Retrospective Analysis
Thank you very much for your kind letter and the excellent comments and for allowing us to submit a revised draft of the manuscript ’’ Sézary Syndrome in West Sweden: Exploring Epidemiology, Clinical Features, and Treatment Patterns in a Registry-Based Retrospective Analysis’’
We appreciate the time and effort you dedicated to providing feedback on our manuscript and are grateful for the insightful comments, which contributed to valuable improvements on our paper. We have incorporated most of the suggestions made by the editor and reviewers. Those changes are marked in red colour.
We hope that the revised manuscript will better suit the Cancers but are also pleased to consider further revisions, and we thank you for your continued interest in our research.
Sincerely,
Karolina Wojewoda/authors
RESPONSE TO REVIEWER 1
In the present study, Wojewoda et al. retrospectively analyzed epidemiologic, clinical and therapeutic data of 17 Sézary Syndrome (SS) patients from a single center in Sweden between 2012 and 2024. It is true that the sample size is low, but the study is important, presenting data regarding SS, the rare and aggressive form of CTCL. Such studies deserve to be published in order to better understand these rare forms of cancers. The manuscript is well written and the results are clear.
Response: We are extremely grateful for your kind and supportive comments! We also want to sincerely appreciate your positive and constructive suggestions, especially for reviewing our work.
Please find below some comments that need to be addressed:
- Weirdly, I did not see any SS patient treated with HDACi (i.e. Vorinostat, Romidepsin) in this study. Is there a specific reason behind that? Or any national regulation (maybe it is not the case, as stated in the limitation section)? Knowing that these drugs are approved for the treatment of SS. And several centers use these drugs for refractory or relapse CTCL.
I guess this idea should be discussed and clearly stated within the manuscript (PMID: 17962618, 16960145, 33229141, and https://doi.org/10.1016/j.fander.2023.03.006 in french). And in both cases, a section talking about HDACi should be added.
Also, these drugs are reported to impact many aspects of cancer cells in CTCL as single agents or in combination (PMID 35408897, 34765562, 28695331, 30529073).
Response: Thank you for raising this excellent point. We understand that HDACi may be proposed in advanced stages of MF or SS in patients who do not respond to at least one prior line of systemic treatment; however, in Sweden, they are not approved and listed in therapeutic guidelines for the management of SS and are also not included in the latest recommendations from Latazka from 2023 (Latzka, J.et al. EORTC consensus recommendations for treating mycosis fungoides/Sézary syndrome – Update 2023. Eur J Cancer 2023, 195, 113343, doi:10.1016/j.ejca.2023.113343).
The treatment with HDACi is an important therapeutic option; we have added this information and new reference (Chebly, A. et al. Diagnosis and treatment of lymphomas in the era of epigenetics. Blood Rev 2021, 48, 100782, doi:10.1016/j.blre.2020.100782) to the discussion now.
The revised text discussion part: in 4.8 Limitations reads as follows: However, due to access and prescribing restrictions, we were not able to use all of the treatments outlined in the latest guidelines from 2023 [42] or treatments with Histone Deacetylase Inhibitors (HDACi) that have shown promise in the treatment of SS [52].
-Also, no genetic data available in the manuscript. Is there any molecular testing performed for these patients? NGS panels? If yes, please also include this information. Or explain. Especially that the detection of genetic alterations is known to impact drug response.
Response: Thank you so much for the comments and for bringing up those questions concerning the genetic data. Yes, genetic profiling can provide valuable insights into the underlying mechanisms of SS; however, we did not conduct a molecular study for genetic profiling of SS cases.
We added the new text in the discussion part: 4.5. Diagnostic Findings that reads as follows: 4.5.4 Molecular Findings. Genetic profiling can provide valuable insights into the underlying mechanisms of SS; however, we did not conduct a molecular study for genetic profiling of SS cases.
-Minor comment: Table 6 is not correctly placed in the text.
Response: We thank the reviewer for this comment. We are sorry that we have not correctly placed Table 6 in the text.
We have now placed Table 6 in 3.4. Disease Progression but even added new text in 3.4. Disease Progression that reads as follows: Table 6 presents clinical outcomes and detailed information for 17 patients with SS.
-Maybe another limitation: absence of data regarding the use of HDACi
Response: Thank you for this recommendation. Data regarding treatment with HDACi is now added in the discussion section.
The revised text discussion part: in 4.8 Limitations reads as follows: However, due to access and prescribing restrictions, we were not able to use all of the treatments outlined in the latest guidelines from 2023 [42] or treatments with Histone Deacetylase Inhibitors (HDACi) that have shown promise in the treatment of SS [52].
Reviewer 2 Report
Comments and Suggestions for Authors
The paper is very well written, and the data is nicely presented. Two weak facts about this paper, 1, the number of cohort is small, only 17 cases. 2. lack of molecular study for genetic profiling of SS cases. A minor issue, the ethnicity of the group was not mentioned. Nevertheless, I think this is a very good paper regarding SS.
Author Response
Cancers (ISSN 2072-6694)
AUTHORS COMMENTS TO THE EDITOR AND REVIEWER
Manuscript ID cancers: 3015464
Wojewoda et al
Title: Sézary Syndrome in West Sweden: Exploring Epidemiology, Clinical
Features, and Treatment Patterns in a Registry-Based Retrospective Analysis
Thank you very much for your kind letter and the excellent comments and for allowing us to submit a revised draft of the manuscript ’’ Sézary Syndrome in West Sweden: Exploring Epidemiology, Clinical Features, and Treatment Patterns in a Registry-Based Retrospective Analysis’’
We appreciate the time and effort you dedicated to providing feedback on our manuscript and are grateful for the insightful comments, which contributed to valuable improvements on our paper. We have incorporated most of the suggestions made by the editor and reviewers. Those changes are marked in red colour.
We hope that the revised manuscript will better suit the Cancers but are also pleased to consider further revisions, and we thank you for your continued interest in our research.
Sincerely,
Karolina Wojewoda/authors
RESPONSE TO REVIEWER 2
The paper is very well written, and the data is nicely presented.
Response: We thank the reviewer for this remark and for taking the time to provide excellent feedback on our manuscript.
Two weak facts about this paper,
1, the number of cohort is small, only 17 cases.
Response: We agree that the number of cohort is small, with only 17 cases – this is an excellent and very helpful suggestion.
As described in 4.8. Limitations, there is currently no national register in Sweden. If such a register were set up, then future studies would be able to utilize data from the entire country, potentially aiding in a larger number of patients. Of course, we will take the comment into account, and we hope that work on the national register will be completed soon.
- lack of molecular study for genetic profiling of SS cases.
Response: Thank you so much for the comments and for bringing up this question concerning the genetic data. Yes, genetic profiling can provide valuable insights into the underlying mechanisms of SS; however, we did not conduct a molecular study for genetic profiling of SS cases.
We added the new text in the discussion part: 4.5. Diagnostic Findings that reads as follows: 4.5.4 Molecular Findings. Genetic profiling can provide valuable insights into the underlying mechanisms of SS; however, we did not conduct a molecular study for genetic profiling of SS cases.
A minor issue, the ethnicity of the group was not mentioned.
Response: Thank you for this recommendation. The new text in Results 3.3.1. Demographic and Clinical Characteristicsreads as follows: All participants had Swedish ethnicity.
Nevertheless, I think this is a very good paper regarding SS.
Response: We are extremely grateful for your kind and supportive comments!
Kind regards,
Karolina Wojewoda/authors